# Current Gallstone Treatment Methods, State of the Art

**DOI:** 10.3390/diseases12090197

**Published:** 2024-08-26

**Authors:** Xiangtian Li, Jun Ouyang, Jingxing Dai

**Affiliations:** 1The Second Clinical Medical College, Southern Medical University, Guangzhou 510280, China; 3208010154@i.smu.edu.cn; 2Guangdong Provincial Key Laboratory of Digital Medicine and Biomechanics, Guangdong Engineering Research Center for Translation of Medical 3D Printing Application, National Virtual, Reality Experimental Education Center for Medical Morphology (Southern Medical University), National Key Discipline of Human Anatomy School of Basic Medical Sciences, Southern Medical University, Guangzhou 510515, China; jouyang@smu.edu.cn

**Keywords:** minimally invasive treatment, percutaneous transhepatic choledochoscopic lithotripsy, percutaneous transhepatic one-step biliary fistulation, endoscopic retrograde cholangiopancreatography, laparoscopic common bile duct exploration and extraction, choledocholithiasis

## Abstract

This study aims to provide valuable references for clinicians in selecting appropriate surgical methods for biliary tract stones based on patient conditions. In this paper, the advantages and disadvantages of various minimally invasive cholelithiasis surgical techniques are systematically summarized and innovative surgical approaches and intelligent stone removal technologies are introduced. The goal is to evaluate and predict future research priorities and development trends in the field of gallstone surgery. In recent years, the incidence of gallstone-related diseases, including cholecystolithiasis and choledocholithiasis, has significantly increased. This surge in cases has prompted the development of several innovative methods for gallstone extraction, with minimally invasive procedures gaining the most popularity. Among these techniques, PTCS, ERCP, and LCBDE have garnered considerable attention, leading to new surgical techniques; however, it must be acknowledged that each surgical method has its unique indications and potential complications. The primary challenge for clinicians is selecting a surgical approach that minimizes patient trauma while reducing the incidence of complications such as pancreatitis and gallbladder cancer and preventing the recurrence of gallstones. The integration of artificial intelligence with stone extraction surgeries offers new opportunities to address this issue. Regarding the need for preoperative preparation for PTCS surgery, we recommend a combined approach of PTBD and PTOBF. For ERCP-based stone extraction, we recommend a small incision of the Oddi sphincter followed by 30 s of balloon dilation as the optimal procedure. If conditions permit, a biliary stent can be placed post-extraction. For the surgical approach of LCBDE, we recommend the transduodenal (TD) approach. Artificial intelligence is involved throughout the entire process of gallstone detection, treatment, and prognosis, and more AI-integrated medical technologies are expected to be applied in the future.

## 1. Introduction

More than twenty years ago, the quality of surgical procedures was primarily evaluated based on speed [1]; however, postoperative retrospective studies [2] have shown that patients often have poor prognoses. This is because overly rapid surgeries do not allow sufficient time to ensure the complete removal of lesions, leading to high recurrence rates [3]. Additionally, large surgical incisions require longer recovery times, increase the risk of infection, and affect cosmetic outcomes. To address these issues, significant advancements have been made in minimally invasive techniques. Over the past decade, precision and minimal trauma have become the new standards for evaluating surgical quality [4]. For example, in the UK, over 70,000 cholecystectomies are performed annually [5], with the majority being completed laparoscopically, provided the patient has no contraindications [6]. Furthermore, with continuous improvements in endoscopic functions and updates in visualization equipment, an increasing number of minimally invasive stone extraction techniques have been developed, offering patients tailored surgical options. The inherent advantages of minimally invasive techniques, such as precision and minimal trauma, have greatly improved patient outcomes since their introduction [7,8,9,10,11,12,13].

This study examines three commonly used and technically advanced minimally invasive gallstone surgery methods. It primarily evaluates and compares the advantages and disadvantages of each method and discusses the new technologies derived from these techniques to assist clinicians in selecting personalized surgical options for their patients. Additionally, the integration of artificial intelligence (AI) with ERCP is considered a future trend in stone surgery, which this study also explores.

## 2. Percutaneous Transhepatic Choledochoscopic Lithotripsy (PTCS)

Traditional methods for treating patients with choledocholithiasis include choledochotomy, partial hepatectomy, and liver transplantation [14]. However, due to the complex anatomical structure of bile duct stones and the low degree of visualization, residual stones cannot be removed [15]. This condition is often accompanied by recurrent postoperative episodes, inflammation induced by repeated stimulation of the bile duct by the stones, a high number of surgical complications, and a high mortality rate [16]. Percutaneous transhepatic choledochoscopic surgery (PTCS) was first reported in 1968 as a new type of lithotripsy (Figure 1) [17]. Now, PTCS utilizes choledochoscopic techniques to extract stones under direct visualization, allowing for higher accuracy, fewer residual stones, and less trauma. Wang et al. [18] found that the recurrence rate is only 9% and that the procedure is highly suitable for refractory ductal calculi.

### 2.1. Long-Term Methods

The traditional two-step percutaneous transhepatic biliary drainage (PTBD) procedure requires the patient to first undergo a percutaneous liver puncture using a needle. This needle must be inserted parallel or slightly oblique to the longitudinal axis of the bile duct. A guide wire is then inserted through the needle into the intrahepatic bile duct, followed by the sequential insertion of dilation tubes along the guide wire in increasing sizes, starting with an 8 F tube. Over approximately four weeks, these tubes are replaced with progressively larger ones until the bile duct is dilated to 12–18 F, allowing for the insertion of a choledochoscope. After dilation is complete, the patient undergoes surgery under general anesthesia, during which the choledochoscope is inserted along the sheath. Depending on the size of the stones, they are either retrieved with a basket or pushed through the duodenal papilla for expulsion. 

### 2.2. Short-Term Methods

For the percutaneous transhepatic one-step biliary fistula (PTOBF) procedure, the patient undergoes general anesthesia before direct percutaneous biliary puncture. Following the insertion of a guide wire through the puncture needle, dilation tubes of increasing sizes are sequentially placed to dilate the tract for the choledochoscope. This method eliminates the waiting time for bile duct dilation required in the long-term methods, instead using dilation tubes to forcefully expand the duct. Once the bile duct is dilated to 14–16 F [2], a rigid choledochoscope is introduced to perform stone extraction. The subsequent steps are the same as in the long-term methods.

The advantage of the PTOBF is the gradual and gentle expansion of the fistula tract, minimizing the risk of bleeding [19,20]. However, because this method is time-consuming, often requiring >1 month from puncture and drainage to final extraction of the stone, it is not applicable to patients in the acute phase [3]. In addition, if the patient has a severe adverse reaction during the extraction, the operation needs to be stopped and the drainage tube put back in place. The tube requires continual drainage for several days before the operation can be continued, which can be an ordeal for both the patient and the surgeon.

Some researchers have shown no statistically significant differences in the stone retrieval rate and postoperative complications (*p* > 0.05) between the PTBD and PTOBF procedures; however, the latter option is associated with a higher rate of bleeding [19,21]. This may be related to the rapid dilation of the fistula or increased pressure caused by the lack of biliary drainage.

Overall, PTOBF is well regarded as an emerging method for extracting stones. In addition to greatly reducing the operating time, this method has two additional advantages. First, the use of sheaths to protect the bile ducts shortens surgical access, facilitates the extraction and flushing out of stones, and reduces stimulation of the biliary tract by the surgical instruments, reducing the patient’s stress reaction. Second, a rigid choledochoscope is more cost-effective and easier to operate than a fiberoptic cholangioscope. The choledochoscope provides more space for the extraction of stone fragments and other procedures and thus improves the efficiency of stone extraction when there are multiple incarcerated stones.

### 2.3. Introduction of a Modified Method For PTCS

Some clinicians have combined the PTBD and PTOBF procedures [22]. One week after puncture and drainage, the patient is put under general or epidural anesthesia, and the sheath tube is delivered into the intrahepatic bile duct with the dilator for one-time fistula dilatation. A rigid choledochoscope is then extended into the bile duct through the sheath tube to extract the stone. This modality combines the advantages of both methods, using the sheath applied by the PTOBF to protect the bile duct and improve stone extraction, draining bile to relieve biliary stasis and avoid high pressure in the biliary tract, reducing the rate of bile leakage, and stabilizing the patient’s coagulation and immune function. This combined method can be completed within approximately one week from the time of puncture to the extraction of the stone.

## 3. Endoscopic Retrograde Cholangiopancreatography (ERCP) Combined with Stone Extraction

Gallstone treatment previously required opening the patient’s abdomen or another body surface following laparoscopic exploration. In 1974, Kawai et al. [23] reported on the first lithotripsy using endoscopic retrograde cholangiopancreatography (ERCP, Figure 2). This method has gradually become one of the mainstream procedures for the treatment of gallstones. It has the advantage of not requiring laparotomy and body surface incision, significantly improving the patient’s aesthetic outcome and allowing for hospital discharge within 3–5 days after the operation, thereby reducing the overall medical cost; however, because ERCP requires high stone morphology and operator skills and can cause irreversible damage to the Oddi sphincter, which in turn triggers intestinal and biliary reflux, the pros and cons of this treatment should be weighed against the needs of the patient.

### 3.1. Endoscopic Sphincterotomy (EST)

An endoscopic sphincterotomy (EST) is performed by inserting a duodenoscope into the duodenum after the patient has received basic anesthesia. Cholangiopancreatography is used to observe the bile ducts and the location, number, and size of the stones in the descending part of the duodenum. The duodenal papilla is then incised, and a guide wire is placed into the bile duct through the incised duodenal papilla and combined with a mesh basket to extract the stone. Re-imaging is performed after the completion of stone extraction to ensure that the stones have been cleanly removed and any nasobiliary drainage is left in place [24].

### 3.2. Endoscopic Balloon Dilatation of the Duodenal Papillary Sphincter (EPBD)

After endoscopic balloon dilatation of the duodenal papillary sphincter (EPBD), the patient is administered basic anesthesia according to the size of the stone and the diameter of the bile duct. Balloons of different diameters are placed along the guidewire into the bile duct, and a contrast agent is injected into the balloon to reach a certain pressure until the sphincter waist completely disappears. Inflation is maintained for 30–120 s [25]. The balloon pressure is then reduced and replaced with a duodenal papilloscope for subsequent stone removal.

Since the sphincter of Oddi is cut, EST allows duodenoscopy and lithotripsy instruments to enter the bile duct relatively easily. This has the advantage of limiting damage to the bile duct and increasing the rate of stone removal. Indeed, 85–96% of choledochal stones can be completely removed through the duodenal papilla [26], reducing the risk of pancreatitis. However, because EST requires open surgery and more technical methods, there is an increased risk of intraoperative bleeding as well as longer operation and hospitalization times. Damage to the Oddi sphincter barrier can also increase the odds that intestinal contents or gas will reflux into the biliary tract, elevating the risk of reflux cholangitis and biliary pneumatosis [27].

In contrast, EPBD does not require the sphincter to be incised, minimizing the risk of damaging the Oddi sphincter. The incidence of bleeding is also reduced, which is beneficial to patients with coagulation disorders. In patients with bile duct variations or abnormal anatomical positions (most often after gastric bypass surgery [28,29,30]), this method provides a wider operating space for efficient stone extraction. The downside to not cutting the sphincter and directly dilating the balloon is that the papilla has to withstand higher compression, which can promote duodenal papilla edema [31]. In addition, the repeated mechanical lithotripsy of large stones, entry and exit of instruments from the papillary muscle, and reduced openness of the papillary muscle can obstruct the pancreatic fluid outflow of small stones, increasing the risk of pancreatitis [32].

While EST has been traditionally used to treat common bile duct stones, clinicians have become increasingly concerned about potential damage to the sphincter of Oddi and the time required for papillary stenosis. EPBD, which is less risky and has fewer complications, is more frequently being used because it does not destroy the anatomical structure of the sphincter of Oddi, is associated with less harm to the patient during both imaging and stone removal, requires fewer days of hospitalization, and is more economical [33,34].

### 3.3. Limited Sphincterotomy Combined with Large-Balloon Dilation (Limited EST Plus EPLBD)

Postoperative pressure testing of both EPBD and EST shows a reduction in the contractility of the papillary muscles. In addition, a single EPBD is often useless for patients with stones >1 cm, and forced expansion of the balloon can induce an acute transmural inflammatory reaction and intramucosal hemorrhage. To address this, endoscopic sphincterotomy combined with large-balloon dilatation (ESLBD) was developed. This method incorporates the advantages of both EPBD and EST by making a small incision in the sphincter of Oddi prior to balloon dilatation, increasing the potential diameter of dilatation and allowing the removal of larger diameter stones without mechanical fragmentation. ESLBD also reduces pressure on the sphincter, which in turn lowers the incidence of postoperative papilledema and pancreatitis. The procedure can be completed in an average of seven minutes and even faster when performed by experienced physicians [35]. A method for restoring the function of the sphincter of Oddi is under development [36,37].

### 3.4. Comparison of Different Stone Removal Methods in ERCP

We systematically reviewed clinical trials related to endoscopic retrograde cholangiopancreatography (ERCP) from 1997 to 2023, meticulously selecting studies that focused on endoscopic papillary balloon dilation (EPBD) and endoscopic sphincterotomy (EST). Subsequently, we conducted a comprehensive statistical analysis of the data on patients with choledocholithiasis and their complications, primarily post-ERCP pancreatitis. The results are summarized in Table 1, with a representative article chosen every 3 to 4 years. Additionally, we have clearly indicated the recommended surgical approach from each study at the end of each row to enhance the completeness and practical utility of the information.

In the patient cohort included in this study, the age distribution predominantly centered around 70 years, with a significantly higher proportion of female patients compared to male patients. This trend is consistent with the findings of previous prospective studies [46,47]. This phenomenon may be attributed to the role of estrogen in women, which enhances bile secretion through the hepatic ERα signaling pathway [48], thereby increasing cholesterol saturation. This process can also impair gallbladder motility, leading to bile stasis and subsequently promoting stone formation. Additionally, significant changes in hormone levels exacerbate the incidence of gallbladder and bile duct stones in pregnant women [49]. It is estimated that at least 40,000 young, healthy women in the United States undergo cholecystectomy each year due to postpartum complications [50], highlighting that pregnancy-related gallbladder disease has become a significant cause of morbidity in this population. Given the numerous restrictions and contraindications associated with surgery during pregnancy, this underscores the urgent need to explore ERCP (endoscopic retrograde cholangiopancreatography)-assisted stone removal as a gentler treatment option.

Firstly, difficult cannulation has long been a major obstacle to successful stone extraction in ERCP (endoscopic retrograde cholangiopancreatography). Numerous studies have shown a negative correlation between cannulation success rates and the risk of stone recurrence and complications [51]. Specifically, a large-scale randomized controlled trial showed that when the number of cannulation attempts reached or exceeded ten, the incidence of post-ERCP pancreatitis (PEP) significantly increased [52]. In light of this, PEP is highlighted in this table as a representative high-risk complication associated with increased cannulation attempts. Furthermore, we compared the outcomes of two different balloon dilation durations (60 s and 300 s) in EPLBD (endoscopic papillary large-balloon dilation) procedures in a 2017 cohort study. The results showed that although the cannulation success rate was 83.7% in the 60 s group, its PEP incidence (15.1%) was significantly higher than that of the 300 s group, which had a single-session stone extraction success rate of 95.2%.

Furthermore, we compared the efficacy of two different balloon dilation times (60 s and 300 s) in EPLBD (endoscopic papillary large-balloon dilation) procedures in a 2017 cohort study. The results showed that although the 60-second group had an 83.7% success rate for cannulation, its incidence of post-ERCP pancreatitis (PEP) (15.1%) was significantly higher than that of the 300 s group, which achieved a 95.2% success rate for stone removal in a single attempt. In recent years, with the continuous advancements in ERCP technology, physicians can utilize personalized scoring systems to select the optimal cannulation strategies [53,54,55,56]. This has not only improved the success rate of cannulation but has also effectively prevented certain complications [57,58]. Additionally, with the progress in lithotripsy technology and the modernization of endoscopic equipment, three studies over the past decade have consistently shown that the initial complete stone removal success rate has stabilized above 80.0%, with an increasing trend each year [34,44,45]. The most recent randomized cohort study reported an initial complete stone removal rate of 98.2% for EPLBD, surpassing the 96.5% success rate of EST within the same group [45].

Given the advantages of EPLBD, such as low invasiveness, ease of operation, and cost-effectiveness, some physicians prefer it as the first-line treatment, using EST as a backup option if EPLBD cannulation fails [59,60,61]; however, this approach may inadvertently increase the number of cannulations, leading to a higher incidence of post-ERCP pancreatitis (PEP), rising from 3.3% to 14.3% [62]. Therefore, we recommend performing a sphincterotomy (SO) prior to EPLBD to improve the cannulation success rate, as this strategy does not increase the risk of post-ERCP pancreatitis (PEP). Additionally, a prospective study innovatively proposed a three-step cannulation method (combining traditional cannulation with guidewire [63], double guidewire [64], and needle knife techniques [65]). Experienced operators using this method achieved a cannulation success rate of up to 99%, providing new strategies and insights for the successful implementation of ERCP procedures [66].

The size and number of stones are typically positively correlated with the stone clearance efficiency [67,68]. This necessitates a longer sphincterotomy incision, a larger balloon size, and a longer balloon dilation time to enlarge the sphincter opening and facilitate effective stone removal [69]. In a 2004 study, the EPLBD (endoscopic papillary large-balloon dilation) group had an average stone diameter of 14.0 mm, while the EST (endoscopic sphincterotomy) group had an average stone diameter of 16.0 mm, the largest sizes recorded in studies over the years [40]; however, the initial surgery success rate was the lowest among the cited studies, at only 70%. This phenomenon can be attributed to two factors: first, the constraints of the experimental design, which required the collection of bile from the common bile duct for biochemical analysis, limiting the use of auxiliary guidewire techniques and necessitating traditional cannulation methods; second, the need to preserve sphincter function and avoid excessive dilation that could lead to permanent biliary sphincter damage [70] and subsequent chronic biliary complications [71]. To expand the opening while preserving sphincter function, researchers attempted to spray isosorbide dinitrate (ISDN) at the major duodenal papilla. The vasodilating [72] and smooth-muscle-relaxing properties of ISDN [73] were utilized to promote the dilation of the bile duct and sphincter. If the balloon dilation pressure required for EPBD exceeded 4 mmHg, the procedure was switched to EST. This approach significantly reduced the overall incidence of post-ERCP pancreatitis (PEP) to 1.6%, while preserving approximately 70% of sphincter function and effectively reducing the recurrence rate of common bile duct stones after surgery [42]; however, the preservation effect on sphincter function varied significantly among individuals, limiting its widespread application. Additionally, while mechanical lithotripsy (ML) is an effective method for managing large stones [74], studies indicate that it may increase the risk of stone recurrence and hepatobiliary complications [75]. A randomized cohort study reported that the frequency of ML use was lower in the EPLBD group compared to the EST group (28.8% vs. 46.2%), suggesting that EPLBD offers greater advantages in terms of cost-effectiveness, operational safety, and convenience [38].

In recent years, a large multicenter, single-blind, randomized controlled trial investigated the optimal balloon dilation time when combining EPLBD with EST for treating common bile duct stones [34]. However, due to limitations in trial costs and sample size, it remains challenging to determine the optimal surgical parameters through long-term, large-scale cohort studies to maximize stone removal efficiency while minimizing recurrence and complication risks [76]. Therefore, finding a balance between sphincter damage and stone clearance efficiency to ensure patient outcomes remains an urgent issue [77]. The following sections will discuss the latest technological advancements addressing this problem.

### 3.5. Introduction of a Modified Method For ERCP

The three main methods for stone removal in combination with ERCP involve entering the common bile duct by disrupting the anatomical structure of the duodenal papilla. This common approach poses a challenge as it can result in damage and functional impairment of the sphincter of Oddi, ultimately leading to a loss in pressure gradient between the biliary tract and the duodenal lumen [78]. The resulting reflux of duodenal fluid containing bacteria and other components may lead to a series of adverse events such as CBD recurrence, biliary stricture, biliary infection, and even biliary cancer.

Placing a self-expanding metal stent (SEMS) in the bile duct to allow for dilation before stone removal is a method that may help reduce damage to the surrounding structures and facilitate smooth passage of the duodenoscope into the bile duct. However, challenges such as inadequate bile drainage post-stent placement, duodenal perforation resulting from stent length or looseness, and complex and time-consuming surgical procedures have been reported [79]. Furthermore, the lack of prospective, comparative, large-scale, and long-term studies means that the impact of stent placement on overall survival function preservation is uncertain. Therefore, a new surgical method known as endoclip papillaplasty (ECPP) was developed to use endoclips for clipping the incised Oddi sphincter in order to enhance the healing process following endoscopic sphincterotomy (EST) [80].

In terms of the specific procedures involved in modified surgery, ERCP remains the initial step, with the insertion of a guidewire into the common bile duct being a key part of the process. This is followed by performing localized endoscopic sphincterotomy (EST) near the major duodenal papilla, and then proceeding with endoscopic papillary balloon dilation (EPBD) in a similar fashion as previously performed [78]. After complete removal of the stone, a stent is placed in the biliary tract to facilitate bile outflow and prevent bile duct damage during subsequent OS clamping. Subsequently, ERCP is repeated, and fluoroscopy-guided placement of the endoclips in a zipper-like manner from proximal to distal ends is performed. The angle and positioning of the endoclips are continuously assessed to ensure proper healing of the OS incision. Follow-up post-surgery revealed that the OS pressure in the majority of patients returned to pre-EST levels. Additionally, a prospective cohort study demonstrated an 8.5% reduction in the postoperative CBD recurrence rate in the ECPP group [78], indicating that ECPP effectively restores OS function and improves patient prognosis.

## 4. Laparoscopic Common Bile Duct Exploration and Extraction (LCBDE)

Since laparoscopic cholecystectomy (LC) was popularized in 1989, laparoscopy methods have greatly expanded, including the emergence of laparoscopic common bile duct exploration and extraction (LCBDE) [81]. This modality is minimally invasive, has low clinician competence requirements, a high success rate, low trauma, fast postoperative recovery, and few complications, and is now commonly used to remove extrahepatic bile duct stones(Figure 3).

### 4.1. The Transcystic (TC) Duct Approach

The transcystic (TC) duct approach, also known as laparoscopic transcystic bile duct exploration and extraction (LTCBDE), involves placing the laparoscope using a standard four-hole approach (i.e., umbilicus, subxiphoid, right midclavicular line 2 cm below the subcostal margins, and right anterior axillary line). After freeing the choledochal duct from its confluence with the common hepatic duct, it is longitudinally dissected or micro-incised against the common bile duct, and a choledochoscope is inserted into the duct via the midclavicular line (forceps can be extended through the port of the xiphoid process to direct its entry into the common bile duct (CBD)). Stone extraction is completed by closing the duct with an absorbable hem-o-lok clip and placing a drainage tube through the hole of Winslow [82].

### 4.2. The Transduodenal (TD) Approach

For the transduodenal (TD) approach [83,84], a laparoscope is placed using the same standard four-hole approach, and a direct incision of the common bile duct is made to remove the stone. The incision is then closed with a 4–0 absorbable suture and a drain is placed in Winslow’s orifice using a T-tube. The drain is removed 5–6 weeks after cholangiography or cholangioscopy with no stone residue [30,85].

The TC duct approach causes less trauma, reduces postoperative morbidity rates, and limits the need for an indwelling T-tube in the postoperative period. This shortens the time needed for recovery and hospitalization, lowers the cost, and does not require a high level of technical competence since there is no need for suturing and knot-tying [86]. However, the TD approach is more effective at extracting large and intrahepatic stones and improves postoperative decompression of the biliary tract due to the retention of the T-tube [87]. This method also limits stenosis and bile duct obstruction, prevents bile outflow responsible for biliary peritonitis [88,89], and leaves a passageway for the subsequent and more convenient removal of any stones that are not easily extracted, reducing the need for an additional operation [90].

While both the TC duct and TD approaches can handle multiple stones and are suitable for patients with mild inflammation, stone diameters of >6 mm, cystic duct diameters of <4 mm, and/or the presence of intrahepatic stones, the TC duct method can have a more negative impact on the patient’s body [82]. There is no statistically significant difference in the stone removal rate of the two approaches; however, the incidence of bile leakage is significantly higher among patients receiving the TC duct approach than the TD method.

The TC duct method eliminates the need for an indwelling T-tube, improving patient recovery time. Indeed, the postoperative morbidity rate is lower for patients receiving the TC duct approach than laparoscopic choledochotomy, partly due to postoperative CBD stenosis resulting from disruption of the choledochal anatomy and insertion of the T-tube [91,92,93]. There is also a positive correlation between indwelling T-tubes and postoperative complications, indicating that placement of the T-tubes is unnecessary [94].

In theory, the TC duct route is the preferred method if it is supported by the patient’s indicators. If this method is initially chosen, conversion to a choledochotomy approach is still possible, even if it proves difficult to remove CBD stones during a later surgery.

### 4.3. Introduction of a Modified Method For LCBDE

Some researchers have modified LCBDE to pursue a “non-invasive” incision route to the CBD. There is some concern that even a micro-incision of the CBD through the TC duct route could increase the risk of CBD stenosis post-surgery. This has led to the adoption of laparoscopic CBD exploration via the confluence of the choledochotomy duct (LTD-CBDE) model. After an incision site is selected, the choledochal duct is circumferentially incised 0.5 cm away from the CBD in a 3/4 cross-section. The confluence is then dilatated with a columnar dilatation balloon to improve choledochoscope access. This method ensures the integrity of the common bile duct, and no patients developed a stricture during follow-up. However, whether this method fully eliminates the risk of CBD stenosis requires additional study [95].

## 5. Summary

This study provides a comprehensive comparison of the application scenarios and advantages of the three surgical methods (PTCS, ERCP, and LCBDE) in the context of treating common bile duct stones. While ERCP and LCBDE are both established techniques for this purpose, ERCP often necessitates an additional surgical procedure to fully treat the stones, resulting in a multi-step process. On the other hand, LCBDE can efficiently handle multiple surgical operations, such as stone and gallbladder removal, in a single operation. Furthermore, the success of ERCP stone removal is influenced by various factors, including stone size and location as well as surgeon expertise, leading to a notable risk of surgical failure. In cases where ERCP fails, PTCS can serve as a viable alternative to stone treatment [96].

In current practice, ERCP is increasingly being favored over LCBDE for treating gallstones, especially asymptomatic CBD stones [97]. This shift may be attributed to the extensive laparoscopic training required for LCBDE, leading to longer training periods and higher costs [98]. Additionally, with the growing elderly population, many individuals are unable to tolerate the adverse effects of anesthesia, surgical trauma, and postoperative infections. Consequently, the less invasive nature of ERCP combined with stone removal is becoming increasingly popular among patients.

## 6. Future Directions

While ERCP presents several challenges—including the use of a radioactive contrast agent, difficulties in intubation due to anatomical variations, and the potential for incomplete stone removal resulting in stone recurrence—it is evident that there is significant potential for enhancement compared to PTCS and LCBDE. The advancements in artificial intelligence in recent years have had a significant impact on ERCP. In the field of gastrointestinal endoscopy, artificial intelligence has been utilized in the treatment of esophageal cancer [99], the screening of early gastric cancer [100], enteroscopies, the identification of colorectal polyps, and other fields [101,102]; however, further research is required to explore the application and clinical integration of artificial intelligence in conjunction with biliary endoscopy. We believe this will be a key development direction for biliary endoscopy in the future.

The benefits of combining artificial intelligence (AI) and ERCP are illustrated in Figure 4. Due to variations in age, pregnancy status, symptoms, and disease duration, not all patients with common bile duct (CBD) issues necessitate therapeutic ERCP. Establishing the criteria for determining when ERCP is truly needed can be challenging to avoid unnecessary medical procedures. Therefore, some researchers have suggested utilizing artificial neural network (ANN) models to predict which patients are suitable candidates for ERCP procedures [103]. Patient information is compiled into a dataset by adjusting the weight of each patient’s physical examination indicators, performing data cleaning, and other necessary steps. This dataset is then used to train the artificial neural network (ANN) model. Furthermore, a computer-assisted (CAD) system has been created to assess, score, and categorize the level of difficulty of stone extraction during ERCP [104]. AI tools enable doctors to better evaluate patients’ surgical risks, prognosis, and treatment options. This allows doctors to select more personalized surgical approaches for patients, as opposed to relying solely on traditional CBD prediction models [105].

Doctors performing ERCP are often exposed to radiation for extended periods due to the need to use fluoroscopy during operations. However, leveraging AI to take over certain tasks traditionally performed by doctors can significantly reduce operation times, subsequently lowering radiation exposure. AI, specifically trained through a convolutional neural network (CNN)-based algorithm, can identify critical anatomical structures in the surgical process and design a virtual surgical approach [106]. Finally, postoperative complications such as acute pancreatitis (AP) greatly impact patient satisfaction and recovery. AP is a common complication of ERCP, with an incidence rate of 3.5–9.7% (up to 14.7% in high-risk patients [107]). To address this problem, a team utilized machine learning (ML) to model the data of patients with CBD stones, aiming to predict the occurrence of postoperative AP. They also employed the SHapley Additive exPlanation (SHAP) method to elucidate the results of AP prediction and the significance of various indicators [108]. This innovative approach enhances prediction accuracy, enabling doctors to tailor treatment options effectively.

The application of artificial intelligence in the management of biliary tract diseases spans from disease onset to diagnosis and treatment. In fact, AI not only aids doctors in evaluating surgical complexity but also enhances surgical success rates, reduces the operation duration, and significantly enhances patient outcomes. Despite these advancements, many studies in this area have yet to be clinically implemented. Thus, there is a pressing need to further explore the integration of AI with medical interventions in biliary endoscopy to benefit a larger population of patients with gallstones.

## 7. Conclusions

First, regarding whether to perform preoperative puncture for PTCS surgery, we recommend a combined approach of PTBD and PTOBF. This method involves preoperative puncture and bile drainage one week before surgery, reducing the risks of bile stasis and high bile duct pressure. Moreover, the use of a rigid choledochoscope facilitates the removal of large stones and is more cost-effective than a fiber choledochoscope.

In addition, for stone extraction based on ERCP, we do not recommend performing EPLBD first and then using EST as a backup for failed cannulation, as this increases the number of cannulations and carries a higher risk of postoperative pancreatitis and other complications. Although there is no statistically significant difference between EPLBD and EST in terms of stone removal efficacy and preservation of the sphincter of Oddi function, EPLBD has received more positive evaluations [109,110,111,112,113]. Based on an analysis of existing studies, we recommend a small sphincterotomy of the Oddi sphincter followed by 30 s of balloon dilation as the optimal ERCP-based stone extraction procedure. If conditions allow, placing new types of biliary stents after complete stone removal can prevent biliary obstruction and the recurrence of stones [114,115,116,117].

Lastly, for the choice of surgical approach in LCBDE, although TC is the gold standard, we recommend the transduodenal (TD) approach. This approach not only provides a broader view for stone removal but also allows for repeated stone extraction through the residual surgical channel in case of short-term recurrence. Additionally, the incidence of bile leakage is lower in patients who choose TD compared to those who choose TC.

Currently, the cutting-edge applications of artificial intelligence in gallstone treatment include using artificial neural networks (ANNs) [103] to determine a patient’s suitability for surgery and developing computer-assisted (CAD) systems [104] to assist doctors in scoring the difficulty of stone extraction and avoiding surgical risks. Virtual surgical approaches are constructed using convolutional neural network (CNN)-based algorithms [105], guiding doctors in stone removal through virtual imaging technology. The SHAP method [108] is used to predict the incidence of postoperative pancreatitis and other complications.

## 8. Advantages

This article analyzes various established stone surgery methods and proposes enhancements, providing a reference for improved surgical options for clinicians. Additionally, through a rigorous literature screening and analysis in chronological order, we statistically compare balloon dilation and sphincterotomy during ERCP across different time intervals (every 3–4 years), documenting diverse recommendations from authors. These findings, presented in Table 1, are unique among current reviews. Finally, this article integrates artificial intelligence to forecast future ERCP application methods, aligning with current developmental trends.

## 9. Limitations

Firstly, while some of the improved methods discussed in this article show promising patient outcomes, their implementation remains limited to a few hospitals. However, the lack of systematic cohort studies has resulted in a dearth of universally accepted surgical evaluation standards. Secondly, variations in sample sizes across studies, as shown in the statistical analysis presented in Table 1, introduce potential biases. Although smaller studies reported no complications, the influence of random factors cannot be dismissed. Therefore, we should determine the optimal conditions for ERCP, i.e., whether to perform a sphincterotomy, the extent of incision, or the timing and duration of balloon dilation, and large-scale, multicenter studies are required to validate these approaches.

## Figures and Tables

**Figure 1 diseases-12-00197-f001:**
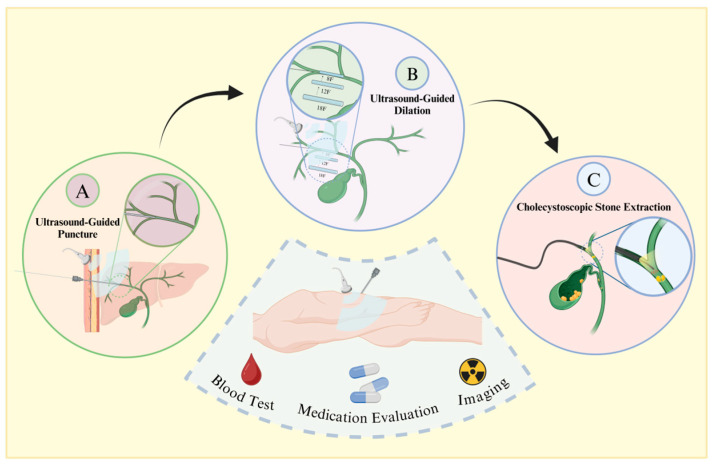
Procedural flowchart of PTCS. Prior to surgery, patients undergo blood assessment, imaging examinations, and medication evaluation. Following confirmation of indications and contraindications, under general anesthesia, the procedure proceeds as follows: A. Using ultrasound guidance, a puncture needle is inserted along the longitudinal axis or slightly tilted into the gallbladder duct, with insertion typically at the left lobe of the liver and the right posterior lobe bile duct. B. Sequential insertion of dilators starting from 8 French size is guided by the wire, ensuring ultrasound guidance to prevent organ damage. C. Post-dilation, a choledochoscope is inserted through the sheath, allowing for mechanical lithotripsy or basket extraction of stones, followed by placement of a drainage tube.

**Figure 2 diseases-12-00197-f002:**
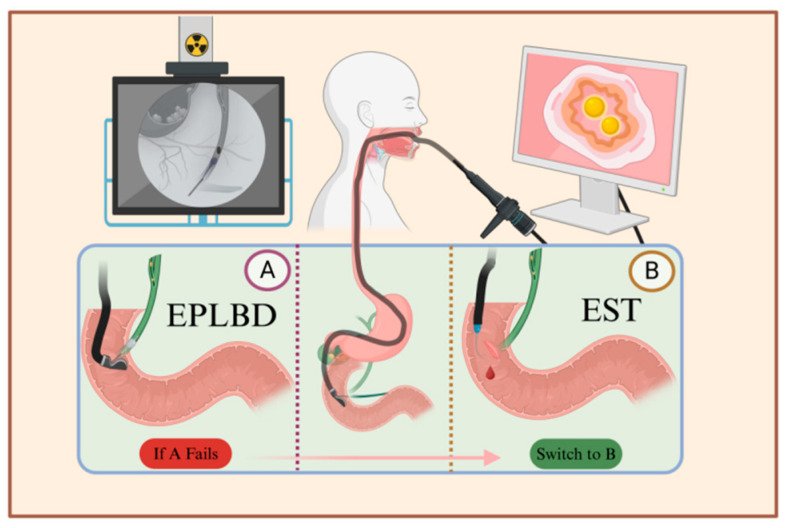
Illustration of the ERCP procedure. After local or general anesthesia, the patient is positioned on their side, and an endoscope is inserted through the mouth to reach the major duodenal papilla. A syringe is then used to inject contrast agent into the bile duct through the sphincter, visualizing the pathway of the bile duct. Depending on the size and location of the stones, either EPLBD or EST is chosen. A. Using the endoscope, a balloon is inserted into the bile duct, and after dilation, a basket is used to extract the stones. B. The sphincter is incised between the 11 o’clock and 12 o’clock positions to facilitate stone removal. If catheterization fails in plan A, plan B is employed, which involves incising the sphincter followed by balloon dilation and stone extraction.

**Figure 3 diseases-12-00197-f003:**
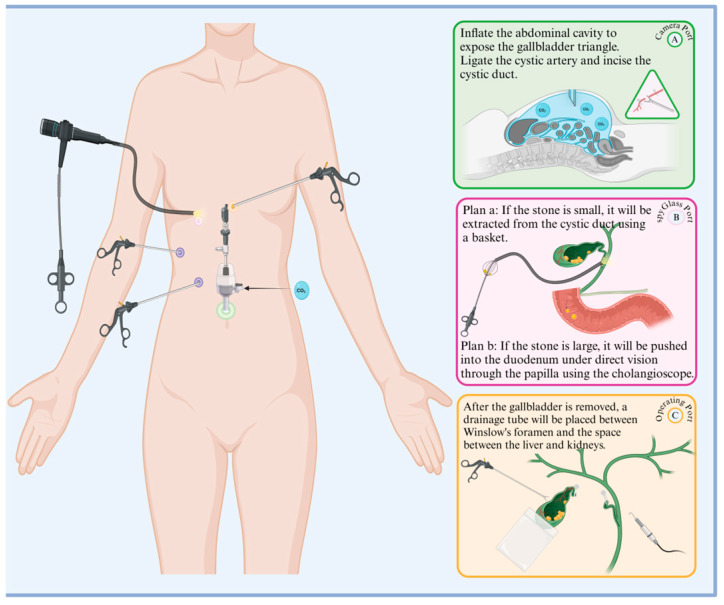
Procedural flowchart of LCBDE. The green, pink, and yellow ports on the left side of the diagram correspond to the right side of the diagram: A. observation port, B. choledochoscope port, C. main operation port, and auxiliary purple ports. The specific operations for each port are listed in sequence in the diagram.

**Figure 4 diseases-12-00197-f004:**
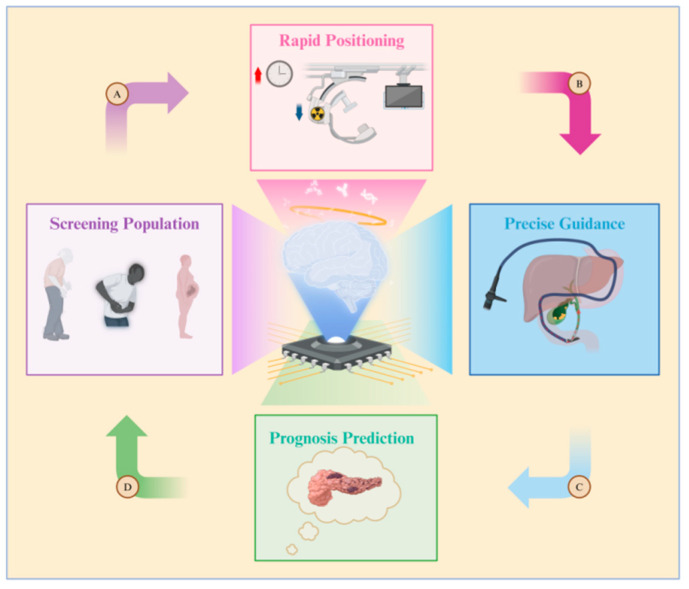
Diagram illustrating the application of artificial intelligence in ERCP. The roles of artificial intelligence are as follows: A. Screening populations at high risk for gallstones, B. Reducing both surgery time and the radiation exposure for doctors and patients, C. Constructing guidance pathways for biliary cannulation, D. Predicting the likelihood of complications such as post-ERCP pancreatitis.

**Table 1 diseases-12-00197-t001:** Comparative analysis of endoscopic balloon dilation (EPBD) versus endoscopic sphincterotomy (EST) in the management of choledocholithiasis across various stages, with recommendations for surgical interventions.

Year	Participants (*n*)	Gender (M/F)	Age (Average)	Complete Stone Removal in First Session	Stone Size (Mean)	Endoscopic Sphincterotomy	Balloon Dilation Time	Post-ERCP Pancreatitis	Recommendation
1997 [38]	101	43/58	72 (29–98)	81%	10.0 mm	0 mm	45–60 s	6.9%	Preserving biliary-sphincter function following EBD may prevent long-term complications.
	101	45/56	71 (29–96)	92%	9.0 mm	Free passage of a fully bowed sphincterotome	0 s	6.9%	
2001 [39]	35	19/16	69.5 (43–86)	N/A	12.7 mm	0 mm	60 s	5.7%	In most cases, EPBD results in a slight reduction in SO function; however, SO function can be preserved to a greater extent compared to EST.
	35	14/21	69.4 (43–88)		13.6 mm	5–10 mm	0 s	5.7%	
2004 [40]	46	32/14	70 (40–90)	70%	14.0 mm	0 mm	60 s	9.5%	Preserving the sphincter of Oddi function may not be essential as a selection criterion when choosing between EPBD or EST.
	45	30/15	69 (41–93)	70%	16.0 mm	Small Medium Large	0 s	8.4%	
2007 [41]	90	51/39	69.1 ± 13.1	72.2%	12.7 mm	0 mm	60 s	16.7%	EPBD appears to be particularly useful in patients with liver cirrhosis who are prone to developing bleeding.
	90	49/41	70.2 ± 8.1	57.8%	11.8 mm	5–10 mm	0 s	6.7%	
2010 [42]	453	261/192	60.9 ± 14.7	N/A	7.7 mm	N/A	60 s	1.7%	The combination of low-pressure EPBD with isosorbide dinitrate preserved papillary function by 70%, potentially enhancing long-term prognosis.
	233	104/129	73.3 ± 13.0		11.1 mm		0 s	0.8%	
2013 [43]	78	40/38	72.97 ± 13.42	88.5%	13.26 mm	Free passage of a fully bowed sphincterotome	0 s	3.8%	EPBD decreased the need for ML and was less expensive.
	73	32/41	71.62 ± 14.8	83.3%	12.47 mm	One-third to one-half of the size of the papilla	30 s	2.7%	
2017 [44]	72	41/31	64.7 ± 15.9	83.7%	6.0 mm	0 mm	60 s	15.1%	Performing EPBD for 5 min is safe and does not increase the risk of recurrent choledocholithiasis.
	80	44/36	61.2 ± 17.4	95.2%	6.3 mm	0 mm	300 s	4.8%	
2019 [34]	371	162/209	65 (IQR)	97%	10.0 mm	3–5 mm	0 s	12%	The recommended approach is a balloon dilation time of 30 s combined with a small sphincterotomy.
	1549	753/796	64 (IQR)	90%		3–5 mm	30, 60, 180 s	10.0%	
2023 [45]	168	84/84	70.1 ± 14.2	98.2%	12.7 mm	0 mm	30 s	4.8%	EPLBD alone was simple, effective, and safe compared with ES-LBD.
	57	20/37	69.6 ± 14.4	96.5%	13.1 mm	One-third to one-half of the size of the papilla	30 s	1.8%	

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
