# Peer review of "Current Gallstone Treatment Methods, State of the Art"

_diseases, 2024, doi:10.3390/diseases12090197_

Round 1
Reviewer 1 Report
Comments and Suggestions for Authors
Dear Dr. Li and Colleagues,
Thank you and I had the pleasure in reading your manuscript related to Various methods incorporated in the management of gallstones.
I recommend using simple diagrams for each method which might make the manuscript more attractive to read and easy to follow.
Best wishes
Comments on the Quality of English Language1. Title: recommend "Current Gallstone Treatment Methods, State of the Art" In the abstract you describe PTCS, ERCP and LCBD in a few lines.
Author Response
Dear Dr. Li and Colleagues,
Thank you and I had the pleasure in reading your manuscript related to Various methods incorporated in the management of gallstones.
Comment 1. I recommend using simple diagrams for each method which might make the manuscript more attractive to read and easy to follow.
Author’s response:
We greatly appreciate your valuable suggestions. After careful consideration, we reviewed available surgical videos and literature, and consulted several hepatobiliary surgery experts in our hospital. We have summarized and outlined the procedures and key points for three minimally invasive stone extraction methods. Additionally, we enlisted a team member with artistic skills to create detailed illustrations. These flowcharts, which are often lacking in most online articles about minimally invasive stone extraction, are one of the highlights of our paper. We confidently believe that these diagrams will be well-suited to your esteemed journal.
Comment 2. Title: recommend "Current Gallstone Treatment Methods, State of the Art" In the abstract you describe PTCS, ERCP and LCBD in a few lines.
Author’s response:
Thank you for your suggestion. We think the title you suggested is better. We have now modified the title to “Current State of the Art in Gallstone Treatment Methods”. Additionally, we have rewritten the abstract and background sections to better align with this title, making the content more concise and relevant. The revised sections have been highlighted for your review.
Reviewer 2 Report
Comments and Suggestions for Authors
The topic is of interest and the review overall well written. My comments:
1) Is the focus only on gallstone or on biliary stones in general? Reading the paper it seems the latter, in spite of the title....
2) The section on ERCP should be expanded. The authors should comment on the evidence on different ERCP treatments for stones (cite the recent NMA: PMID: 34666153 and PMID: 34543649 )
3) Some tables would be useful to improve the paper
Author Response
The topic is of interest and the review overall well written.
Comment 1.Is the focus only on gallstone or on biliary stones in general? Reading the paper it seems the latter, in spite of the title....
Author’s response:
First, we sincerely appreciate your recognition. Your encouragement is our motivation for continuous improvement. Your understanding is indeed correct. In fact, the surgical approaches mentioned in our paper are applicable to both gallstones and biliary stones in general. However, in this article, we focus more on the latter. While common gallstones have well-established surgical treatments, choledocholithiasis (bile duct stones) urgently requires the development of more minimally invasive surgical techniques that can be performed in a smaller operative field. These techniques are of greater reference value to clinicians. We may have caused some confusion due to unclear expressions in certain sections of the article. Therefore, we have revised the parts that may have led to misunderstandings and highlighted these modifications in the section headings.
Comment 2. The section on ERCP should be expanded. The authors should comment on the evidence on different ERCP treatments for stones (cite the recent NMA: PMID: 34666153 and PMID: 34543649 ) Some tables would be useful to improve the paper.
Author’s response:
Thank you very much for your suggestion. The references you recommended have been extremely helpful. We also believe that a high-quality table is an essential addition to our article, in addition to the figures we have included. Therefore, during the week we spent revising the manuscript, we had numerous discussions and, following your recommended references, created a table comparing patient data for two ERCP surgical methods. Additionally, we innovatively expanded on the table content from your recommended references by providing a horizontal comparison of ELPBD and EST, allowing readers to more clearly see the advantages and disadvantages of both procedures. This is a feature not commonly addressed in most ERCP-related clinical trials. We have also added section 2.4 to provide a detailed explanation and analysis of the table content. Your suggestion has greatly enhanced our article, and we express our sincere gratitude once again.
Reviewer 3 Report
Comments and Suggestions for Authors
The article lack of adequate references in most of its parts.
Concepts are presented without adequate logic, especially 1.1 and 2.1
English level is very low
Comments on the Quality of English LanguageIt is difficult to understand adequately the meaning due to the low level of English
Author Response
Comment 1. The article lack of adequate references in most of its parts.
Author’s response:
Thank you for your corrections. First, we agree with your opinion that there were some inaccuracies. Therefore, we have re-reviewed the content cited from other sources in the article and, based on existing historical literature and several new references, we have inserted dozens of additional references to make our article more rigorous(All newly inserted references have been highlighted in the text). Second, some surgical details mentioned in the article were obtained through consultations with several hepatobiliary surgery experts. As a result, they may lack internationally recognized surgical standards. However, we have still inserted corresponding references for universally accepted surgical steps.
Comment 2. Concepts are presented without adequate logic, especially 1.1 and 2.1
English level is very low. It is difficult to understand adequately the meaning due to the low level of English.
Author’s response: Thank you for your evaluation of our article. We greatly value your suggestions. Upon receiving your feedback, we immediately held a meeting to review the entire article. Through team discussions, we identified several areas where the logic was unclear. We also consulted with several gallstone surgery experts at our hospital to discuss the widely used minimally invasive gallstone surgery procedures. Based on their expert guidance and several references, we revised and rewrote sections, including but not limited to, 1.1 and 2.1. Following the recommendations of another reviewer, we added several procedural diagrams for minimally invasive stone extraction surgeries to enhance the clarity of the article's logic. Regarding the issue of language proficiency, we had a native English-speaking medical professional from our team spend about a week polishing the entire content to meet your standards. Thank you for your corrections!
Round 2
Reviewer 2 Report
Comments and Suggestions for Authors
The authors consistently improved their manuscript. Only a minor point:
i recommend to replace ref 63 with a more comprehensive SRMA that is the only network meta-analysis on the topic: PMID: 34543649)
Author Response
Comment 1. The authors consistently improved their manuscript. Only a minor point: i recommend to replace ref 63 with a more comprehensive SRMA that is the only network meta-analysis on the topic: PMID: 34543649)
Author’s response:
We sincerely appreciate your suggestion. We have replaced ref 63 with the literature you recommended. We are deeply impressed by your ability to help us identify more suitable references with such precision. Your specific and detailed feedback has significantly reduced our revision time. If this article is successfully published, your assistance will have played a crucial role. Once again, we extend our heartfelt thanks.
Reviewer 3 Report
Comments and Suggestions for Authors
Substantial modifications were made but the scientific pertinence of this study is still very low.
Comments on the Quality of English LanguageMinor revisions are still necessary
Author Response
Comment 1. Substantial modifications were made but the scientific pertinence of this study is still very low.
Author’s response:
Thank you for your suggestion. Regarding the issue of scientific pertinence that you mentioned, we have provided a more concise description in the conclusion of the abstract. Additionally, in the main body of the article, we have offered an objective and comprehensive discussion of various minimally invasive gallstone removal surgeries. We have also listed the most appropriate surgical methods based on the findings from several large cohort studies, which is content not found in other articles on minimally invasive gallstone surgeries. We believe this will greatly aid clinicians in their surgical choices and improve patient outcomes. Moreover, by comparing the structural shortcomings of other publications, we have refined the discussion and outlook of this paper, offering valuable insights for the future development of this field. We have also highlighted the limitations of our study, which will guide our future research. Lastly, we have outlined many other strengths of this article in the conclusion. In summary, these are the adjustments we made in response to your concerns about scientific pertinence.
Comment 1. Minor revisions are still necessary.
Author’s response:
We sincerely accept your critique and have made logical adjustments to the content related to the selection of surgical procedures without altering the overall meaning of the article. This was done to enhance the clarity for readers and to further improve the rigor of the paper. Additionally, we compared our work with other similar articles and found that most of them lack well-designed figures and tables. As a result, we streamlined the surgical details in the figures and updated some of the references to more advanced and systematic studies, aiming to provide clinicians with the most up-to-date surgical techniques. Finally, please be assured that, after extensive discussions and revisions by our team, the scientific rigor, practicality, and presentation quality of this article now surpass those of similar works. We once again appreciate your constructive criticism.
Round 3
Reviewer 3 Report
Comments and Suggestions for Authors
Adequate revisions made.
Comments on the Quality of English LanguageAdequate revisions made.
Author Response
Comments 1: Adequate revisions made.
Respone 1: The article has undergone English language editing by MDPI. The text has been checked for correct use of grammar and common technical terms, and edited to a level suitable for reporting research in a scholarly journal. MDPI uses experienced, native English speaking editors.
